# A quantitative approach for the analysis of clinician recognition of acute respiratory distress syndrome using electronic health record data

**Meagan A. Bechel**[1,2], **Adam R. Pah**[3,4], **Hanyu Shi**[5], **Sanjay Mehrotra**[6], **Stephen D. Persell**[7,8], **Shayna Weiner**[9], **Richard G. Wunderink**[10], **Luís A. Nunes Amaral**[3,5,11], **Curtis H. Weiss**[12]*

**1** Medical Scientist Training Program, Feinberg School of Medicine, Northwestern University, Chicago, IL, United States of America, **2** Department of Biomedical Engineering, Northwestern University, Evanston, IL, United States of America, **3** Northwestern Institute on Complex Systems, Northwestern University, Evanston, IL, United States of America, **4** Kellogg School of Management, Northwestern University, Evanston, IL, United States of America, **5** Department of Chemical and Biological Engineering, Northwestern University, Evanston, IL, United States of America, **6** Department of Industrial Engineering and Management Sciences, Northwestern University, Evanston, IL, United States of America, **7** Division of General Internal Medicine and Geriatrics, Feinberg School of Medicine, Northwestern University, Chicago, IL, United States of America, **8** Center for Primary Care Innovation, Institute for Public Health and Medicine, Feinberg School of Medicine, Northwestern University, Chicago, IL, United States of America, **9** Michigan Oncology Quality Consortium, Ann Arbor, MI, United States of America, **10** Division of Pulmonary and Critical Care Medicine, Feinberg School of Medicine, Northwestern University, Chicago, IL, United States of America, **11** Department of Physics and Astronomy, Northwestern University, Evanston, IL, United States of America, **12** Division of Pulmonary, Critical Care, Allergy, and Immunology, NorthShore University HealthSystem, Evanston, IL, United States of America

* CWeiss@northshore.org

**Data Availability Statement:** The fully identified data set cannot be shared publicly because it contains HIPAA Protected Health Information. A

## Abstract

### Importance

Despite its efficacy, low tidal volume ventilation (LTVV) remains severely underutilized for patients with acute respiratory distress syndrome (ARDS). Physician under-recognition of ARDS is a significant barrier to LTVV use. We propose a computational method that addresses some of the limitations of the current approaches to automated measurement of whether ARDS is recognized by physicians.

### Objective

To quantify patient and physician factors affecting physicians' tidal volume selection and to build a computational model of physician recognition of ARDS that accounts for these factors.

### Design, setting, and participants

In this cross-sectional study, electronic health record data were collected for 361 ARDS patients and 388 non-ARDS hypoxemic (control) patients in nine adult intensive care units at four hospitals between June 24 and December 31, 2013.

de-identified dataset containing all data required to reproduce our findings is available on a public repository managed by Northwestern University: https://doi.org/10.21985/n2-33my-2s89. Requests for the full data set including dates of admission, intubation, mortality, and full ventilator data will require approval of the Northwestern Institutional Review Board and should be submitted to the corresponding author (CWeiss@Northshore.org). Full data will be made available for researchers who meet the criteria for access to confidential data as set forth by the Northwestern Institutional Review Board (https://irb.northwestern.edu/).

**Funding:** This project was supported by the National Institute of General Medical Sciences, Grant T32GM008152 (MB); the National Heart, Lung, and Blood Institute, Grant K23HL118139 and Grant R01HL140362-01A1 (CHW); the Francis Family Foundation (Parker B. Francis Fellowship Program, CHW); the Department of Defense Army Research Office, Grant W911NF-14-1-0259 (MB, LANA, CHW); the National Center for Research Resources, Grant 5UL1RR025741, which is now at the National Center for Advancing Translational Sciences, Grant 8UL1TR000150 (Northwestern University Clinical and Translational Sciences Institute Enterprise Data Warehouse); and John and Leslie McQuown (LANA). The content is solely the responsibility of the authors and does not necessarily represent the official views of the National Institutes of Health, the Francis Family Foundation, or the Department of Defense. The funders had no role in study design, data collection and analysis, decision to publish, or preparation of the manuscript

**Competing interests:** SP reports grant support from Pfizer, Inc. unrelated to this manuscript. MB, LANA, and CHW report the related US provisional patent: Systems and Methods for Patient Management Within a Healthcare Facility. US Serial No: 62/457,574.The other authors declare that they have no conflicts of interest. This does not alter our adherence to PLOS ONE policies on sharing data and materials.

## Methods

Standardized tidal volumes (mL/kg predicted body weight) were chosen as a proxy for physician decision-making behavior. Using data-science approaches, we quantified the effect of eight factors (six severity of illness, two physician behaviors) on selected standardized tidal volumes in ARDS and control patients. Significant factors were incorporated in computational behavioral models of physician recognition of ARDS.

## Results

Hypoxemia severity and ARDS documentation in physicians' notes were associated with lower standardized tidal volumes in the ARDS cohort. Greater patient height was associated with lower standardized tidal volumes (which is already normalized for height) in both ARDS and control patients. The recognition model yielded a mean (99% confidence interval) physician recognition of ARDS of 22% (9%-42%) for mild, 34% (19%-49%) for moderate, and 67% (41%-100%) for severe ARDS.

## Conclusions and relevance

In this study, patient characteristics and physician behaviors were demonstrated to be associated with differences in ventilator management in both ARDS and control patients. Our model of physician ARDS recognition measurement accounts for these clinical variables, providing an electronic approach that moves beyond relying on chart documentation or resource intensive approaches.

## Introduction

Despite a broad consensus on the virtues of translating evidence into clinical practice, adoption of evidence-based practices remains slow.[1,2] The use of low tidal volume ventilation (LTVV) for the treatment of acute respiratory distress syndrome (ARDS) is a prime example. ARDS is a syndrome of severe acute hypoxemia and non-cardiogenic inflammatory lung injury with high prevalence (10% of intensive care unit (ICU) admissions) and mortality (35–46%).[3–6] In the 2000 ARDS Network trial, it was shown that lowering the volume of each breath–i.e. using LTVV—is an effective therapy for ARDS, with a relative mortality reduction of 20–25%[7]; since then, LTVV has become recommended practice for the management of ARDS. [8] Despite this substantial evidence, LTVV use in clinical practice remains as low as 19%.[9–17] Several studies have examined the barriers to LTVV use, including the primary role of physician under-recognition of ARDS.[3,6,10,12–20] However, the identification of these barriers has not led to a substantial increase in LTVV use, suggesting a need for further investigation.

A hallmark of healthcare quality improvement is the consistent measurement of an outcome (ex: number of infections, checklist use, etc.). In the case of LTVV use for ARDS, measurement of an outcome is challenging for multiple reasons. First, delivering LTVV is a two-step process comprised of i) recognizing ARDS and ii) selecting and adjusting tidal volumes based on patient response. Both steps in this process can be affected by patient characteristics. [3,13,17,20–22] Second, while previous studies have employed LTVV use or physician documentation of ARDS as surrogates for physician recognition of ARDS, these proxies have

limitations. The gold standard for measuring physician recognition of ARDS would be to directly ask physicians if a patient has ARDS. While this approach has been used in the past,[3] it introduces important biases such as the observer effect, subjective reporting, and priming clinicians to think about ARDS.[23–25] Furthermore, it is labor and resource intense, making it an infeasible solution for widespread implementation. An alternative method is to collect physician documentation of ARDS from electronic health records (EHR). While this approach is more scalable, physicians may recognize ARDS but not document it in their notes, nor deliver LTVV despite this recognition.

We sought to use data science approaches on EHR data to address the above challenges with measurement of physician recognition of ARDS and build an estimate of physician recognition of ARDS that could be widely implemented. We use tidal volume selection as a proxy for physician decision-making behavior and quantify the factors affecting tidal volume selection in both ARDS patients and a novel control cohort of patients who do not have ARDS. We then build two models of physician recognition of ARDS that account for these factors. Our methods not only address the issues of bias and resource availability, but also provide insights into underlying physician behavior that have implications for effective intervention design.

## Materials and methods

### Cohort development

We have previously described the development of the ARDS cohort used in this study, which includes 362 patients who met the Berlin Definition of ARDS[5] (see summary below) via independent clinician review and were admitted to an ICU at one academic and three community hospitals in the Chicago region between June 24, 2013 and December 31, 2013.[16]

Berlin Definition Criteria Summary [5]

1. Hypoxemia: $P_aO_2/F_IO_2 \leq 300$

2. Positive chest imaging: bilateral infiltrates (radiological signs in both lungs)

3. Presence of a known risk factor for ARDS

4. Respiratory failure not fully explained by cardiac failure or fluid overload

Abbreviations: $P_aO_2$ –partial pressure of arterial oxygen; $F_IO_2$ –fraction of inspired oxygen

For this study, we developed an additional cohort from the same time period and initial screening population at two of the same hospitals (one academic, one community): 388 patients with acute hypoxemic respiratory failure requiring mechanical ventilation with at least one instance of $P_aO_2/F_IO_2 \leq 300$ but not with ARDS according to the above Berlin Definition ("control cohort"). We excluded patients with missing key information (predicted body weight [PBW], gender, tidal volumes), intubation duration less than 5.67 hours (the shortest duration of intubation in the ARDS cohort), and PBW less than 25 kg (Fig 1).

Patients were not actively recruited for either cohort, but instead all data was mined from the electronic health record. The ARDS and control cohorts were similar across several clinical and demographic measures (S1 Table). These cohorts are representative of the larger population of patients with ARDS and non-ARDS acute hypoxemic respiratory failure due to our broad inclusion criteria, and their similarity to larger cohorts (e.g., LUNG SAFE[3]) with respect to height, weight, and hypoxemia severity. This study was approved by the Northwestern University Institutional Review Board (STU00208049) with a waiver of consent on October 30, 2018.

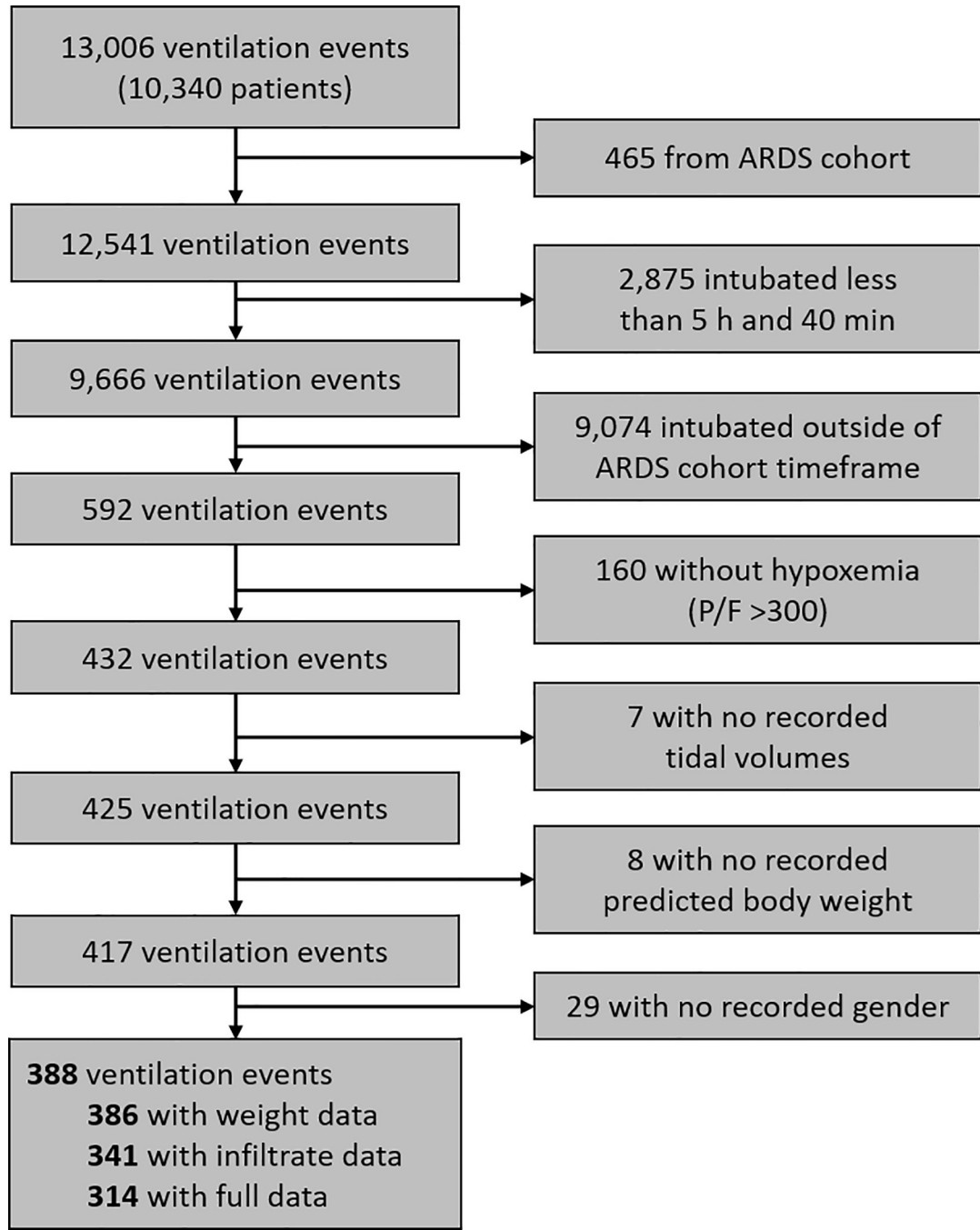

**Fig 1. Flow of patient screening and enrollment for control cohort.**

## Data acquisition

All patient data was obtained from the electronic health records serving the participating hospitals. We defined study entry as the start of ARDS for the ARDS cohort and the first instance of $P_aO_2/F_IO_2 \leq 300$ in the control cohort. Study end was defined as the earlier of extubation, death, or ICU discharge. We recorded gender, height, and all $P_aO_2/F_IO_2$ and weights between ICU admission and study end. We recorded all tidal volumes ($V_T$) and plateau pressures ($P_{plat}$)

between intubation and study end where available ($P_{Plat}$ was not recorded at two hospitals). Note that 22% of the ARDS cohort and 44% of the control cohort only had one unique tidal volume over their study duration. We recorded which ICU the patient was treated in and whether an ARDS diagnosis was documented in the critical care physician's notes. For the control cohort, we recorded whether or not bilateral infiltrates were present for all chest radiographs or computed tomography scans between intubation and study end.[16] For data availability for both cohorts and all subgroups, see S2 Table. Patients who met cohort inclusion criteria but were missing other data points were only excluded from analyses that required those missing data points.

In this study, we use PBW as a gender-adjusted and gender-neutral measurement of height because LTVV thresholds are defined using PBW. Any references to patient height refer to PBW (kg) and any references to patient weight refer to a patient's weight measured at ICU admission (kg). We calculated PBW according to the ARDS Network definition (see below) and defined LTVV as a standardized tidal volume ($\hat{V}_T$) $\leq 6.5$ mL/kg PBW.

Predicted Body Weight Equations [7]:

$$\text{Male}: \quad PBW\ (kg) = 50 + 2.3 * (height(in) - 60)$$

$$\text{Female}: \quad PBW\ (kg) = 45.5 + 2.3 * (height(in) - 60)$$

### Significance testing

We used $\alpha = 0.01$ instead of 0.05 to ensure the statistical strength of our findings [26] and applied the Bonferroni correction for multiple hypotheses. In the regression analyses (see Results), there were 33 comparisons where $\hat{V}_T$ was the dependent variable, thus we set $p < 0.0003$ (0.01/33) as the threshold for statistical significance for these analyses. For the covariate analyses, the threshold was $p < 0.005$ (0.01/2). For the Kolmogorov–Smirnov tests in Model Approach #2, the threshold was $p < 0.003$ (0.01/3).

## Results

### Analysis of potential factors in tidal volume selection

**Factors assessed.** We used the lowest standardized tidal volume ($\hat{V}_T$) (mL/kg PBW) for each patient as the dependent variable in both univariable and multivariable ordinary least squares (OLS) regressions. $\hat{V}_T$ was used as a continuous variable. OLS regressions were implemented using the *statsmodels* (version 0.6.1) Python package.

We determined the relationship between several factors and $\hat{V}_T$, choosing variables that have been identified previously in the literature as potential barriers or facilitators of LTVV use[11,18–21,27]: first $P_aO_2/F_IO_2 \leq 300$, lowest $P_aO_2/F_IO_2$, highest $P_{plat}$, patient weight at ICU admission, ARDS documentation in the patient chart, presence of bilateral infiltrates (control only), admitting ICU (ARDS only), and patient height (we used the gender neutral PBW). These factors comprise measures of illness severity ($P_aO_2/F_IO_2$, $P_{plat}$, radiographic findings), patient characteristics (height, weight), and physician behaviors (ARDS documentation, patient weight). Plateau pressure was included due to the previously reported practice of physicians not lowering tidal volumes in ARDS patients when $P_{plat} \leq 30$ cm $H_2O$.[19,22] Patient weight was included due to the previously reported barrier of physicians using actual body weight instead of predicted body weight in the LTVV threshold calculation [19]. Note that we use a standardized tidal volume ($\hat{V}_T$) as opposed to the recorded tidal volume ($V_T$) and PBW is included as a control variable. Since $\hat{V}_T$ is already normalized for PBW, we expected no

**Table 1. Predictors of lowest standardized tidal volume (mL/kg PBW) (β-coefficient [99% CI]).**

| Factor | ARDS | | Control | Pooled Documented |
|---|---|---|---|---|
| | univariable | multivariable | univariable | univariable |
| Predicted body weight | **-3.8**[*] [-4.7, -2.8] | **-3.7**[*] [-4.8, -2.7] | **-5.1**[*] [-6.0, -4.1] | **-3.2**[*] [-5.2, -1.2] |
| $P_aO_2/F_IO_2$ ratio (lowest) | **1.3**[*] [0.4, 2.4] | 1.1 [0.3, 1.9] | 0.8 [-0.1, 1.6] | 2.3 [-0.3, 4.9] |
| Documentation | **-1.3**[*] [-1.9, -0.6] | **-1.2**[*] [-1.9, -0.6] | -1.2 [-2.3, -0.2] | |
| $P_aO_2/F_IO_2$ ratio (first) | 0.7 [-0.19, 1.5] | | 0.3 [-0.6, 1.1] | 1.2 [-0.9, 3.2] |
| $P_{plat}$ (highest) | -2.2 [-4.1, -0.3] | | -1.5 [-3.2, 0.1] | -3.5 [-6.0, -1.0] |
| ICU admission weight | 0.12 [-1.8, 2.0] | | -0.4 [-1.7, 0.9] | -0.6 [-3.3, 2.2] |
| Bilateral infiltrates** | | | -0.5 [-0.9, 0.0] | |
| Admitting ICU | -0.9 [-1.9, 0.1] | | | |

[*] p < 0.0003

** At least once after hypoxemia onset.

Empty cells indicate category was not used due to data being unavailable or not relevant.

additional remaining relationship between PBW and $\hat{V}_T$. Input variables were rescaled between 0 and 1 to allow for comparison of coefficients.

**Univariable analysis.** The relationship between each factor and $\hat{V}_T$ was investigated through univariable OLS regressions for the ARDS and control cohorts (Table 1). Standardized tidal volume ($\hat{V}_T$) decreased with worsening hypoxemia (lower $P_aO_2/F_IO_2$) and documentation in the ARDS cohort ($p < 0.0003$), but not in the control cohort (Fig 2, Table 1). In both cohorts, $\hat{V}_T$ decreased with increasing PBW (gender neutral height, $p < 0.0003$, Fig 3 and Table 1)–a surprising result since $\hat{V}_T$ already takes PBW into consideration. Plateau pressure, weight at ICU admission, $P_aO_2/F_IO_2$ at study start, admitting ICU (ARDS cohort only), and the presence of bilateral infiltrates (control cohort only) were not associated with significant changes in standardized tidal volume in any cohort or subgroup (Table 1, Fig 4).

**Covariate analysis.** The factors demonstrating a significant association with $\hat{V}_T$ in the univariable analyses were evaluated for covariance with each other using OLS regression. Three factors were evaluated for covariance: PBW, lowest $P_aO_2/F_IO_2$, and documentation of ARDS. PBW was not associated with increasing documentation probability (Fig 3) in both cohorts, which was anticipated. Documentation and lowest $P_aO_2/F_IO_2$ were significantly correlated ($p < 0.005$) in both cohorts (Fig 2). This association was also anticipated as sicker patients are easier to recognize. To test the strength of the documentation and lowest $P_aO_2/F_IO_2$ association, we repeated the univariable analysis on the three major subgroups (ARDS non-documented, control non-documented, and pooled documented) (Fig 4). Only PBW was associated with lower $\hat{V}_T$ in all three subgroups (Table 1, S3 Table). There was no association between PBW and lowest $P_aO_2/F_IO_2$ in both cohorts.

**Multivariable analysis.** Significant factors from the univariable analyses were included in multivariable regressions comprised of all possible linear combinations of the factors and appropriate interaction terms (see S1 Methods). The "best" multivariable model was selected using the Akaike and Bayesian Information Criterion (AIC, BIC).

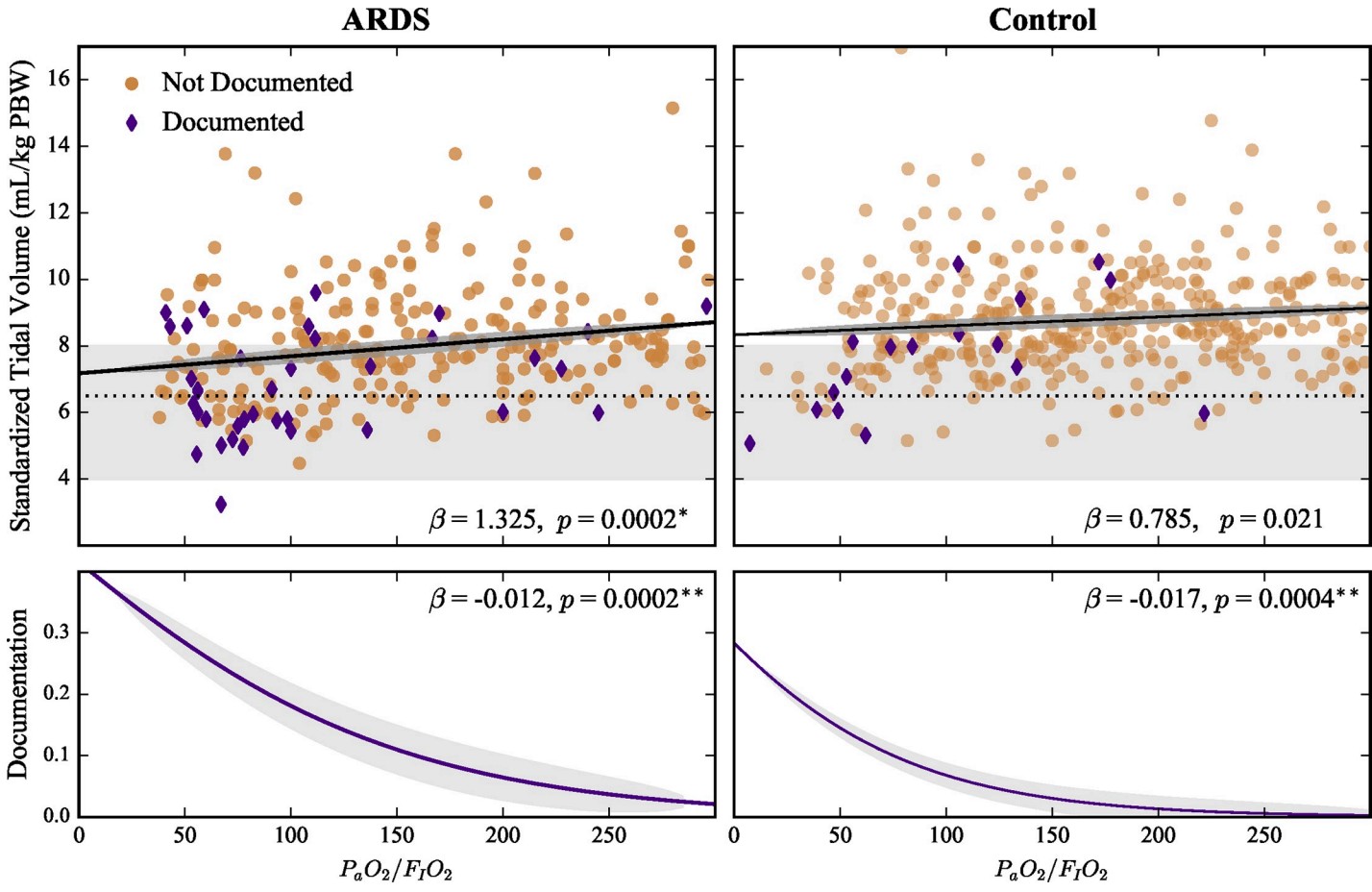

**Fig 2. Effects of lowest $P_aO_2/F_IO_2$ ratio on standardized tidal volume ($\hat{V}_T$) and ARDS documentation in ARDS and control cohorts.** Top panels show patients with ARDS documented in their chart (purple diamonds) and non-documented patients (tan circles). Gray areas indicate LTVV range from current guidelines[7], with dashed line at 6.5 mL/kg PBW from currently recommended threshold. Solid lines show linear ($\hat{V}_T$) and logistic (documentation) fits for scatter plot data (shaded regions, 95% confidence bands). Reported beta coefficients are for standardized inputs. $^*$ $p < 0.0003$, $^{**}$ $p < 0.0005$.

In the ARDS cohort, the multivariable regression model that included PBW, lowest $P_aO_2/F_IO_2$, and documentation as independent variables with no interaction terms resulted in the lowest AIC and BIC (S4 Table). In this model, PBW and documentation were significantly correlated with $\hat{V}_T$ ($p < 0.0003$), while lowest $P_aO_2/F_IO_2$ was not. Of these variables, PBW had the greatest effect on $\hat{V}_T$ (β -3.7, 99% CI -4.8 – -2.7). For the control cohort, only PBW was associated with $\hat{V}_T$, and therefore no multivariable analysis was performed.

**Sensitivity analyses.** To test the robustness of our cohort definitions, we conducted two sensitivity analyses: 1) patients with a study duration longer than 12 hours, and 2) patients within the 2.5–97.5 percentiles of PBW. The first sensitivity cohort is intended to capture clinician behavior, which may require longer time scales, such as a shift change and/or patient rounds. The second sensitivity cohort aimed to evaluate a potential disproportionate effect of PBW outliers on linear trends. Neither sensitivity analyses yielded any difference in the regression results.

## Construction of model of and estimation of physician recognition of ARDS

Sixty patients—44 (12.2%) in the ARDS cohort and 16 (4.2%) in the control cohort—had a documented diagnosis of ARDS by at least one physician. While documentation of ARDS

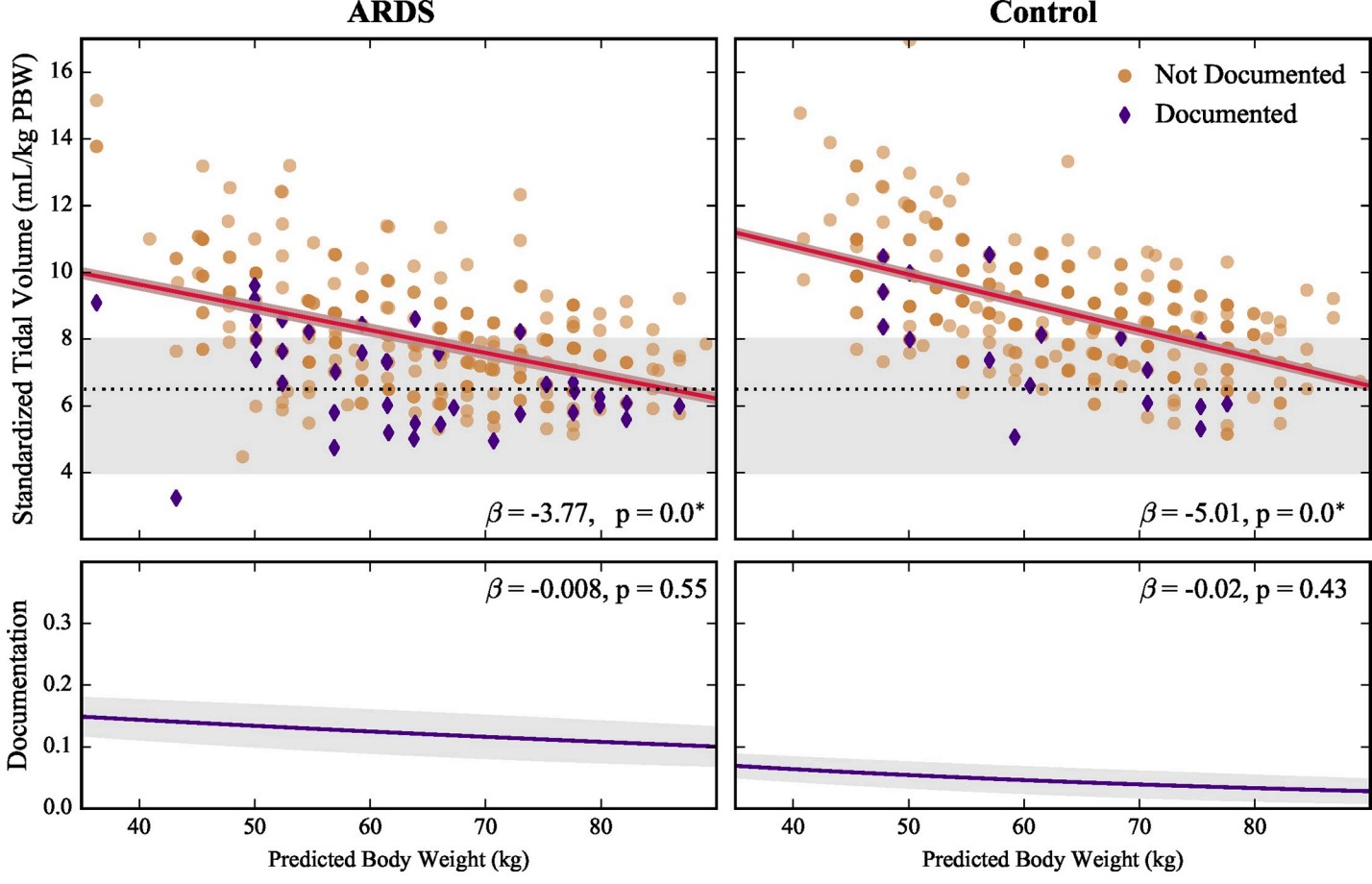

**Fig 3. Effects of predicted body weight (gender neutral height) on standardized tidal volume ($\hat{V}_T$) and ARDS documentation in ARDS and control cohorts.** Top panels show patients with ARDS documented in their chart (purple diamonds) and non-documented patients (tan circles). Gray areas represent LTVV range from current guidelines[7], with dashed line at 6.5 mL/kg PBW at current recommended threshold. Solid lines show linear ($\hat{V}_T$) and logistic (documentation) fits for scatter plot data (shaded regions, 95% confidence bands). Reported beta coefficients are for standardized inputs. * $p < 0.0003$.

implies that the physician recognized ARDS, lack of documentation does not imply that a physician did not recognize ARDS. Beyond their mere documentation practices, a physician's clinical management behavior (e.g., selecting a tidal volume) may shed additional light on whether a physician believes that a patient has ARDS.

We used two approaches to more completely characterize physician recognition of ARDS. To this end, we split the two main cohorts (ARDS and control) into three major subgroups: 1) ARDS non-documented (n = 317), 2) Control non-documented (n = 371), and 3) Pooled documented (n = 60, Fig 4). All patients in the pooled documented cohort have a physician-documented diagnosis of ARDS in their chart. All patients in the non-documented cohorts do not have a physician-documented diagnosis of ARDS in their chart. Our two approaches are based on the assumption that the physician behaviors observed in the ARDS non-documented subgroup represent a mixture of patient-care scenarios in which patients are either recognized by their physician as having ARDS or not recognized as having ARDS. If a patient in the ARDS non-documented subgroup was recognized by their physician as having ARDS, we assume the physician tidal volume selection would be the same as the tidal volume selection seen in the pooled documented subgroup. If a patient in the ARDS non-documented subgroup was not recognized as having ARDS, we assume the physician tidal volume selection would be the

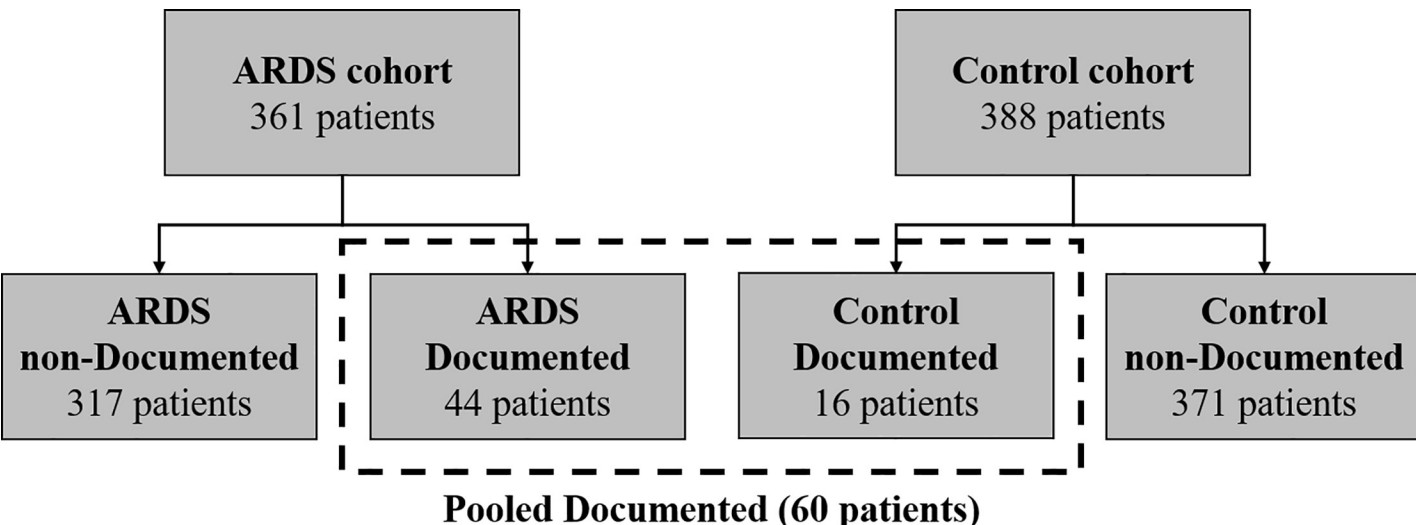

**Fig 4. Cohort and subgroup definitions.**

same as for control non-documented subgroup. Therefore, the non-documented ARDS subgroup patients can be viewed as a mixture of the pooled documented subgroup and the non-documented control subgroup.

**Approach #1: Naïve Bayes.** We used a Naïve Bayes model for classifying patients in the non-documented ARDS subgroup as either recognized or unrecognized by their care teams. We used multivariate kernel density estimation (KDE) to characterize the PBW vs $\hat{V}_T$ clusters for the pooled documented and non-documented control subgroups (Fig 5, top panels). Classifying a patient in the non-documented ARDS subgroup as recognized or unrecognized was based on the following conditional probabilities leveraging Bayes Theorem:

$$\frac{P(documented|PBW, \hat{V}_T)}{P(control|PBW, \hat{V}_T)} = \frac{P(PBW, \hat{V}_T|documented) * P(documented)}{P(PBW, \hat{V}_T|control) * P(control)}$$

In the absence of a reasonable prior for *P(documented)* and *P(control)*, we assign each term 0.5, assuming equal probability of belonging or not belonging to each subgroup. We were able to define a boundary in the PBW vs $\hat{V}_T$ space where *P(documented | PBW, $\hat{V}_T$)* = *P(control | PBW, $\hat{V}_T$)* (Fig 5, top panels, black line). Below this boundary, *P(documented | PBW, $\hat{V}_T$)* is greater than *P(control | PBW, $\hat{V}_T$)* and the patient was classified as 'recognized'. Above this boundary, *P(documented | PBW, $\hat{V}_T$)* is less than *P(control | PBW, $\hat{V}_T$)* and the patient was classified as 'unrecognized'. Due to the size discrepancy between the non-documented control and pooled documented subgroups, we bootstrapped (100 iterations) the "non-documented" control subgroup and repeated this analysis to produce confidence bands (S1 Fig).

The KDE clusters for the pooled documented and control non-documented subgroups as well as the estimated probability equality line are shown in Fig 5. The peaks of male and female PBW (gender neutral height) frequency (Fig 5, bottom panel) align with the two peaks in the pooled documented subgroup (Fig 5, middle panel). Physician recognition of ARDS calculated for each ARDS severity category was: mild, 26%; moderate, 32%; severe, 57% (Table 2).

**Model #2: Mixture Model.** In the second model, we incorporate $\hat{V}_T$, hypoxemia severity (lowest $P_aO_2/F_IO_2$), and PBW with the goal of calculating the fraction of recognized patients

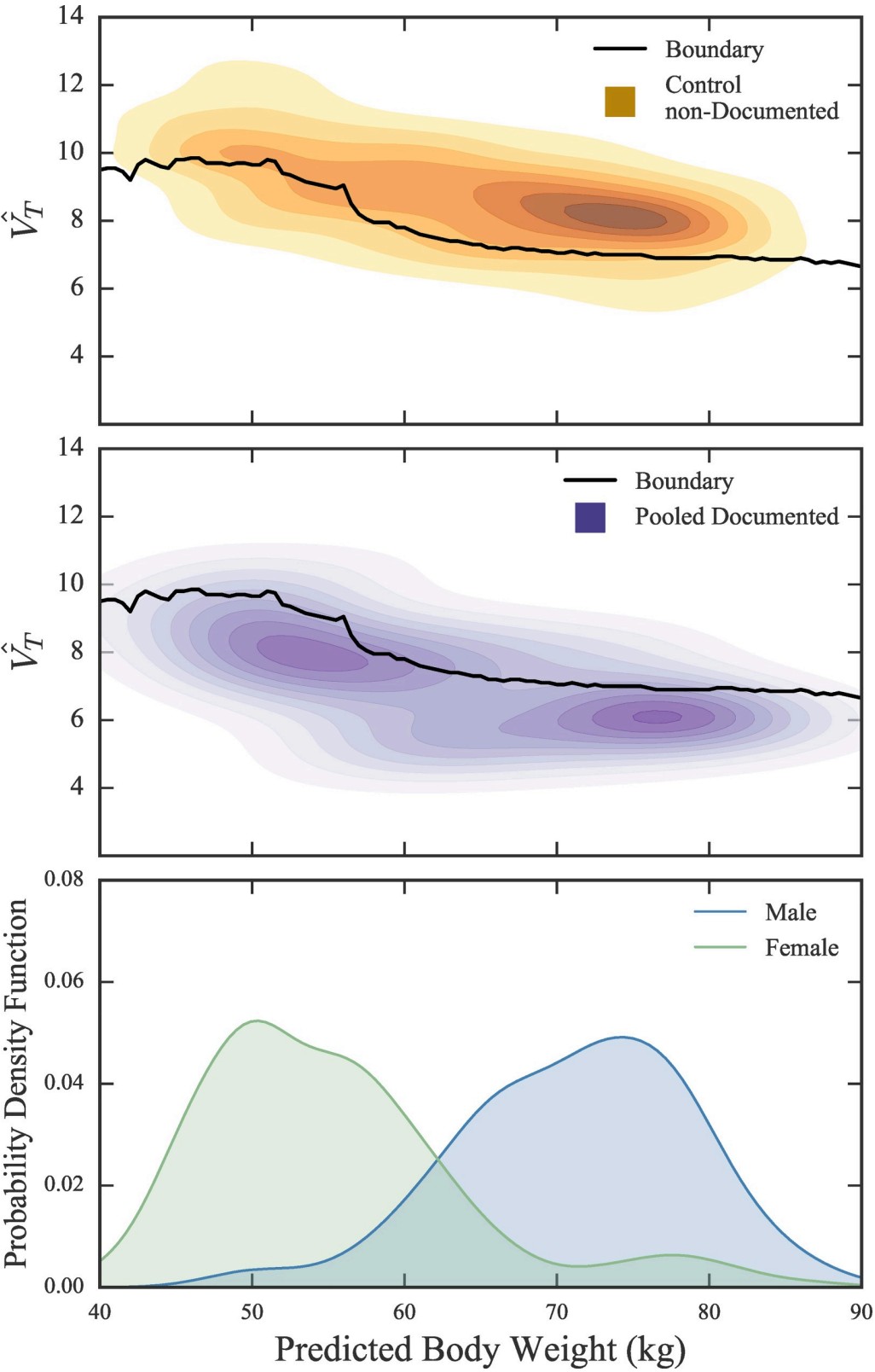

**Fig 5. Kernel Density Estimation for control non-documented and pooled documented patients.** Heatmaps of kernel density estimated probability density for data from control non-documented (yellow, top panel) and documented (purple, middle panel) subgroups. Solid line shows boundary separating region with unequal probability of belonging to

documented (below line) and non-documented control (above line). (Bottom panel) Normalized gender frequency across PBW for combined patient population of documented and control non-documented. Male and female peaks align with high density regions in above heatmaps.

in each Berlin Definition ARDS severity category (mild, moderate, and severe).[5] To calculate physician recognition of ARDS, we estimated the fraction of patients recognized by physicians in each severity category ($f^i_{recognition}$) from the following set of equations:

$$P_{ARDS}(\hat{V}_T, PBW\ data|severity)$$
$$= f^i_{recognition}P_{diagnosed\ ARDS}(\hat{V}_T, PBW\ data) + (1 - f^i_{recognition})P_{non-ARDS}(\hat{V}_T, PBW\ data|severity)$$

where *severity* can take the values "mild," "moderate," or "severe," as set forth in the Berlin Definition [5] (Table 2) and we defined the difference between the probability density functions as the L1 norm:

$$\Delta = \sum |P_{ARDS} - f_i * P_{diagnosed\ ARDS} + (1 - f_i)\ P_{non-ARDS}|$$

where the sum extends over all bins for values of $\hat{V}_T$ and PBW.

We determined the optimal value of $f^i_{recognition}$ by minimizing $\Delta$. Since the corresponding optimization problem is formulated as a linear programming problem, we used *CPLEX* (version 12) as a solver. To determine the uncertainty in our estimates of $f^i_{recognition}$, we used bootstrapping to generate 1000 samples for $P_{ARDS}$ ($\hat{V}_T$, PBW | severity) and repeated the optimization for the bootstrapped samples. As a result, we generated distributions for the optimal value of $f^i_{recognition}$ for each hypoxemia severity category and tested the null hypothesis that these data were drawn from the same distributions with a Kolmogorov–Smirnov test (Python package *scikit-learn* (version 0.18.1)).

This approach yielded a mean (99% confidence interval) physician recognition of ARDS: 22% (9%-42%) for mild; 34% (19%-49%) for moderate; and 67% (41%-100%) for severe (Table 2, S2 Fig). All three recognition distributions were significantly different from each other ($p < 0.003$) when compared via a Kolmogorov–Smirnov test.

## Discussion

We quantified the potential impact of patient characteristics and physician behaviors on the decision-making behavior for tidal volume selection by physicians for patients with ARDS and a novel control cohort. This quantification allowed for the construction of a model to measure physician recognition of ARDS that is not confounded by these patient characteristics and clinical factors. These analyses have allowed us to establish several important findings.

First, we corroborated prior studies' findings that height, hypoxemia severity, and ARDS documentation are associated with the use of lower tidal volumes in ARDS patients.[3,17,19–21,27] We found no evidence for an association between other clinical factors—such as plateau pressure or patient weight—and lower tidal volume use, which have been identified as potential barriers to LTVV use in prior studies.[11,17,19,21] These barriers may still have an impact at the level of the individual physician, but the lack of generalizability to the entire physician population makes them suboptimal for future intervention targets.

Second, our analyses provide additional insight into the previously established relationships between patient height and LTVV use.[21,27] The most common lowest $V_T$ reported in the ARDS and control cohorts were identical (450, 500, and 600 mL), and constitute 51% and 63% of the tidal volumes for the ARDS and control cohorts, respectively. This prevalence of a small

number of lowest $V_T$ suggests that clinicians are not following the canonical relationship between height and lung size originally established in animal studies[28], but instead use a simpler heuristic based on where the patient falls on the height spectrum of their particular gender. This theory is supported by the idea that humans select fast and frugal heuristics under time and knowledge limitations [29], which would both be present in clinical medicine and heightened in critical care. The utilization of this heuristic would translate to a general use of a lower standardized tidal volume ($\hat{V}_T$) for taller patients that is closer to or, in some cases, below the LTVV threshold; which would lead to our observation of the strong relationship between PBW and $\hat{V}_T$, despite that $\hat{V}_T$ already includes PBW in its calculation. Our findings are strong evidence that at least some delivery of LTVV may be unintentional—i.e., solely of a default $V_T$ (450, 500, or 600 mL)—and not based on ARDS recognition or other clinical decision-making factors. While evidence for this physician behavior phenomenon has been previously reported in ARDS patient cohorts [17,27], our findings observe this behavior in a diverse control cohort, implying that the simpler tidal volume selection heuristic use is not restricted to ARDS patients alone.

Alternative explanations for the association between height and LTVV use in both cohorts include some physicians believing in LTVV for patients in the control cohort or some of those patients being classified by physicians as having ARDS. Supporting the latter possibility, 4.2% of control cohort patients had a physician-documented diagnosis of ARDS. Nonetheless, these alternative explanations are less likely because of the low ARDS documentation rate, low use of LTVV in both cohorts, and the strong correlation between PBW (gender neutral height) and $\hat{V}_T$ in both cohorts. Another alternative explanation is physicians using a non-linear relationship between tidal volume and PBW, but this is less likely given the low variability in chosen tidal volumes in both cohorts. Our results suggest that the relationship between PBW (gender neutral height) and $\hat{V}_T$ should be accounted for when measuring LTVV use and when designing implementation strategies to improve LTVV use.

Third, our estimate of physician recognition of ARDS for severe ARDS was comparable to previous studies [3], while our estimated physician recognition of ARDS rates for mild and moderate ARDS were lower. We believe our estimated rates are more plausibly representative of real-world practices. Unlike prior studies, the estimated physician recognition of ARDS rates in our study do not rely on subjective reporting [18,19], the observer effect [3,9,10,13–15], or imposed interventions such as additional training of physicians in ARDS recognition or LTVV use.[3] Our results suggest that the potential impact of these biases was limited to mild and moderate ARDS and was neutralized by the presence of severe ARDS. This finding has implications for the selection of implementation strategies: while the mere presence of severe hypoxemia may be enough to trigger physician recognition of ARDS, additional prompting or training is required to improve recognition of mild or moderate ARDS.

**Table 2. Rates of physician recognition of ARDS by hypoxemia severity.**

| Severity | ARDS Documented | | Recognition | | |
|---|---|---|---|---|---|
| | | | Approach #1: | Approach #2: | LUNG SAFE study [3] |
| | n | % | Naïve Bayes (%) | Mixture Model (% [99% CI]) | (% [95% CI]) |
| Mild $200 < P_aO_2/F_IO_2 \leq 300$ | 5 | 6 | 26 | 22 [9, 42] | 51.3 [47.5, 55.5] |
| Moderate $100 < P_aO_2/F_IO_2 \leq 200$ | 8 | 7 | 32 | 34 [19, 49] | 65.3 [62.4, 68.1] |
| Severe $P_aO_2/F_IO_2 \leq 100$ | 24 | 30 | 57 | 67 [41, 100] | 78.5 [74.8, 81.8] |

## Limitations

Our study has several limitations. First, it was conducted in a single metropolitan area, so we were unable to address regional or national differences. Second, we were limited to the patient data recorded in the EHR, which may be overlooked by physicians in lieu of other information, such as a visual estimation of height. [30] Third, we did not evaluate physician knowledge of ARDS or LTVV, specifically the Berlin criteria and what standardized tidal volume threshold they believe qualifies as LTVV. Alternative LTVV thresholds may be justified by the layout of the ARDS Network tidal volume table, which appears to suggest that tidal volumes ranging from 4 to 8 mL/kg PBW qualify as LTVV.[5] Finally, we acknowledge that it is possible that our application of the Berlin definition may have been biased, leading to misclassification of ARDS or control status—this could also explain why some patients classified as control were documented by their physicians as having ARDS.

## Conclusions

Our findings could have implications for the design of implementation strategies to improve LTVV use. First, we believe documentation and physician recognition of ARDS should be unlinked. Whereas previous studies rely on documentation of ARDS as the *sine qua non* of physician recognition of ARDS [11,13,16,20], there are likely to be patients whose physicians recognized ARDS but did not document it. Our study demonstrates two novel approaches for estimating physician recognition of ARDS that consider additional behaviors beyond documentation (e.g., tidal volume selection), providing a more complete characterization of physician recognition of ARDS. Implementation strategies—which commonly rely on behavior change—should account for the multiple facets involved in physician recognition of ARDS, and multiple channels of data required to measure recognition.

Second, our approach provides a measurement of physician recognition of ARDS that is not subject to confounding by patient characteristics and clinical factors. This approach could be integrated into EHR systems to evaluate an arbitrary number of physicians and sites, which will allow for the comparison of physician recognition of ARDS not only between individuals and institutions, but also over different points in time. This methodology could be used to accurately drive interventions, like clinical decision-support or feedback, to create an even stronger structure for improving the adoption of evidence-based practices.[31,32]

This study offers a compelling example of how data science methods can use EHR data to provide new opportunities for measuring and addressing quality of patient care, specifically in a complex setting such as critical care.[33] While traditional implementation studies allow a broad analysis of the situation as a whole, they are constrained by high costs, logistical complexity, and potential for bias. Our approach achieved similar, and potentially more representative results, while minimizing these disadvantages. Furthermore, our methods may be implementable and sustainable on the local level, providing individual institutions the opportunity to continuously assess or track evidence-based medicine implementation.

## Supporting information

**S1 Methods. Multivariable model selection.**
(DOCX)

**S1 Table. Characteristics of cohorts and subgroups.**
(TIF)

**S2 Table. Data availability for cohorts and subgroups.**
(TIF)

**S3 Table. Predictors of lowest standardized tidal volume (mL/kg PBW) in non-documented subgroups (β-coefficient [99% CI]).**
(TIF)

**S4 Table. Multivariable models of lowest standardized tidal volume (mL/kg PBW) in ARDS cohort.**
(TIF)

**S1 Fig. Naïve Bayes boundary between recognized and unrecognized regions with 95% confidence intervals from bootstrapping.** Scatter plot shows pooled documented patients (purple diamonds) and control non-documented patients (tan circles). Size of marker represents number of data points. Solid line shows boundary separating region with unequal probability of belonging to documented (below line) and non-documented control (above line) with 95% confidence bands from bootstrapped data (shaded region).
(TIF)

**S2 Fig. Distributions of $f^{i}_{recognition}$ from CPLEX analysis of bootstrapped data.**
(TIF)

## Author Contributions

**Conceptualization:** Meagan A. Bechel, Adam R. Pah, Stephen D. Persell, Richard G. Wunderink, Luís A. Nunes Amaral, Curtis H. Weiss.

**Data curation:** Meagan A. Bechel, Shayna Weiner, Luís A. Nunes Amaral, Curtis H. Weiss.

**Formal analysis:** Meagan A. Bechel, Adam R. Pah, Hanyu Shi, Sanjay Mehrotra, Shayna Weiner, Luís A. Nunes Amaral, Curtis H. Weiss.

**Funding acquisition:** Luís A. Nunes Amaral, Curtis H. Weiss.

**Investigation:** Meagan A. Bechel, Adam R. Pah, Hanyu Shi, Luís A. Nunes Amaral, Curtis H. Weiss.

**Methodology:** Meagan A. Bechel, Adam R. Pah, Hanyu Shi, Sanjay Mehrotra, Stephen D. Persell, Luís A. Nunes Amaral, Curtis H. Weiss.

**Project administration:** Meagan A. Bechel, Shayna Weiner, Richard G. Wunderink, Luís A. Nunes Amaral, Curtis H. Weiss.

**Resources:** Luís A. Nunes Amaral, Curtis H. Weiss.

**Supervision:** Adam R. Pah, Stephen D. Persell, Richard G. Wunderink, Luís A. Nunes Amaral, Curtis H. Weiss.

**Validation:** Meagan A. Bechel, Luís A. Nunes Amaral, Curtis H. Weiss.

**Visualization:** Meagan A. Bechel, Luís A. Nunes Amaral, Curtis H. Weiss.

**Writing – original draft:** Meagan A. Bechel, Adam R. Pah, Luís A. Nunes Amaral, Curtis H. Weiss.

**Writing – review & editing:** Meagan A. Bechel, Adam R. Pah, Sanjay Mehrotra, Stephen D. Persell, Shayna Weiner, Richard G. Wunderink, Luís A. Nunes Amaral, Curtis H. Weiss.

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
