## [Decision Letter · Decision Letter 0]

27 Jul 2019

PONE-D-19-17952

A quantitative approach for the analysis of clinician recognition of acute respiratory distress syndrome using electronic health record data

PLOS ONE

Dear Mrs. Bechel,

Thank you for submitting your manuscript to PLOS ONE. After careful consideration, we feel that it has merit but does not fully meet PLOS ONE’s publication criteria as it currently stands. Therefore, we invite you to submit a revised version of the manuscript that addresses the points raised during the review process.

We would appreciate receiving your revised manuscript by Sep 10 2019 11:59PM. To enhance the reproducibility of your results, we recommend that if applicable you deposit your laboratory protocols in protocols.io, where a protocol can be assigned its own identifier (DOI) such that it can be cited independently in the future. For instructions see: http://journals.plos.org/plosone/s/submission-guidelines#loc-laboratory-protocols

We look forward to receiving your revised manuscript.

Kind regards,

Lars-Peter Kamolz, M.D., Ph.D., M.Sc.

Academic Editor

PLOS ONE

Journal Requirements:

1. In your Methods section, please provide additional information about the participant recruitment method and the demographic details of your participants. Please ensure you have provided sufficient details to replicate the analyses such as: a) the recruitment date range (month and year), b) a description of any inclusion/exclusion criteria that were applied to participant recruitment, c) a table of relevant demographic details, d) a statement as to whether your sample can be considered representative of a larger population, e) a description of how participants were recruited, and f) descriptions of where participants were recruited and where the research took place.

2. Thank you for including your competing interests statement; "

SP reports grant support from Pfizer, Inc. unrelated to this manuscript.

MB, LANA, and CHW report the related US provisional patent: Systems and Methods for Patient Management Within a Healthcare Facility. US Serial No: 62/457,574.

The other authors declare that they have no conflicts of interest."

Reviewers' comments:

Reviewer's Responses to Questions

**Comments to the Author**

1. Is the manuscript technically sound, and do the data support the conclusions?

Reviewer #1: Partly

2. Has the statistical analysis been performed appropriately and rigorously? 

Reviewer #1: I Don't Know

3. Have the authors made all data underlying the findings in their manuscript fully available?

Reviewer #1: No

4. Is the manuscript presented in an intelligible fashion and written in standard English?

Reviewer #1: No

5. Review Comments to the Author

Reviewer #1: Dear authors,

Thank you for the opportunity to review the manuscript, “A quantitative approach for the analysis of clinician recognition of acute respiratory distress syndrome using electronic health record data”

A few remarks:

Possibly, the already listed Limitations could receive the title, “Limitations.”

I suggest starting with the conclusion section with line 354.

PlosOne’s required citation style is to be used correctly (e.g. line 350, 339)

The references have to be double-checked. The literature has to be updated.

Hardly any current literature is cited.

What are the actual and above all current results of this study, which would ultimately justify a publication in a scientific journal?

The date of the Ethics Statement is missing.

6. PLOS authors have the option to publish the peer review history of their article (what does this mean?). If published, this will include your full peer review and any attached files.

Reviewer #1: No

---

## [Author Response · Author response to Decision Letter 0]

21 Aug 2019

We have carefully considered all comments and revised our manuscript accordingly. Below please find our point-by-point response to all editor and reviewer comments. We believe these revisions strengthen our manuscript and we hope that it is now suitable for publication in PLOS ONE.

Thank you for your consideration of our manuscript.

Editor Comments

Editor Comment 1: In your Methods section, please provide additional information about the participant recruitment method and the demographic details of your participants. Please ensure you have provided sufficient details to replicate the analyses such as: a) the recruitment date range (month and year), b) a description of any inclusion/exclusion criteria that were applied to participant recruitment, c) a table of relevant demographic details, d) a statement as to whether your sample can be considered representative of a larger population, e) a description of how participants were recruited, and f) descriptions of where participants were recruited and where the research took place.

Response: We have added S1 Table: "Characteristics of cohorts and subgroups,” to provide relevant demographic data. 

We have also revised the methods to include all of the above details, including the following statement:

“Patients were not actively recruited for either cohort, but instead all data was mined from the electronic health record. The ARDS and control cohorts were similar across several clinical and demographic measures (S2 Table). These cohorts are representative of the larger population of patients with ARDS and non-ARDS acute hypoxemic respiratory failure due to our broad inclusion criteria, and their similarity to larger cohorts (e.g., LUNG SAFE) with respect to height, weight, and hypoxemia severity. [3]”

Editor comment 2: Thank you for including your competing interests statement; "SP reports grant support from Pfizer, Inc. unrelated to this manuscript. MB, LANA, and CHW report the related US provisional patent: Systems and Methods for Patient Management Within a Healthcare Facility. US Serial No: 62/457,574. The other authors declare that they have no conflicts of interest." Please confirm that this does not alter your adherence to all PLOS ONE policies on sharing data and materials, by including the following statement: "This does not alter our adherence to PLOS ONE policies on sharing data and materials.” (as detailed online in our guide for authors http://journals.plos.org/plosone/s/competing-interests). If there are restrictions on sharing of data and/or materials, please state these. Please note that we cannot proceed with consideration of your article until this information has been declared.

Response: Below is our updated Conflict of Interest statement:

"SP reports grant support from Pfizer, Inc. unrelated to this manuscript. MB, LANA, and CHW report the related US provisional patent: Systems and Methods for Patient Management Within a Healthcare Facility. US Serial No: 62/457,574. The other authors declare that they have no conflicts of interest. This does not alter our adherence to PLOS ONE policies on sharing data and materials.”

Editor comment 3: a) If there are ethical or legal restrictions on sharing a de-identified data set, please explain them in detail (e.g., data contain potentially identifying or sensitive patient information) and who has imposed them (e.g., an ethics committee). Please also provide contact information for a data access committee, ethics committee, or other institutional body to which data requests may be sent.

b) If there are no restrictions, please upload the minimal anonymized data set necessary to replicate your study findings as either Supporting Information files or to a stable, public repository and provide us with the relevant URLs, DOIs, or accession numbers.

Response: After further consultation with our IRB, we have confirmed that we are approved to host a fully de-identified data set on a public repository. We have provided a copy of our data at: https://doi.org/10.21985/n2-33my-2s89

Reviewer Comments

Reviewer Comment 1. Possibly, the already listed Limitations could receive the title, “Limitations.”

Response: We agree that this would improve the structure of the paper and have edited the manuscript accordingly.

Reviewer Comment 2. I suggest starting with the conclusion section with line 354.

Response: We have updated the conclusion section to start where suggested.

Reviewer Comment 3: PlosOne’s required citation style is to be used correctly (e.g. line 350, 339)

Response: The citations have been updated to meet the required style, including the two lines cited.

Reviewer Comment 4: The references have to be double-checked. The literature has to be updated. Hardly any current literature is cited.

Response: We agree that current literature is an important aspect to include. With an eye toward more recent relevant literature, we have added the following three additional citations that focus on ARDS recognition and implementation of appropriate ventilator strategies. 

• Spece LJ, Mitchell KH, Caldwell ES, Gundel SJ, Jolley SE, Hough CL. Rate of Low Tidal Volume Ventilation Use Remains Low in Patients with Acute Respiratory Distress Syndrome Despite Improvement Efforts at a Single Center. J Crit Care. 2018 Apr; 44: 72-76.

• Owyang CG, Kim JL, Loo G, Ranginwala S, Mathews KS. The effect of emergency department crowding on lung-protective ventilation utilization for critically ill patients. J Crit Care. 2019; 52: 40-47.

• Matthay MA, Zemans RL, Zimmerman GA, Arabi YM, Beitler JR, Mercat A, Herridge M, Randolph AG, Calfee CS. Acute respiratory distress syndrome. Nat Rev Dis Primers. 2019 Mar 14; 5 (1):18. 

The first study provides a confirmation that the low utilization issues demonstrated in our cohort affect other large academic hospitals. This specific study takes place in a former ARDSnet trial hospital, demonstrating adoption drop-off after the clinical trial and the need for a more sustainable implementation strategy.

The second study confirms our tidal volume frequency findings of 450 and 500 mL as well as the lack of tidal volume changes for a single patient. Furthermore, it proposes default values within electronic orders as a source, which supports our conclusion that default tidal volumes may be the source of relationship between patient height and low tidal volume ventilation use. 

The third study is a Nature Review Disease Primer, which provides details on the specific challenges of diagnosing ARDS and how that may affect treatment choices.

Reviewer Comment 5: What are the actual and above all current results of this study, which would ultimately justify a publication in a scientific journal?

Response: The main results of this study are twofold. First, we expand on the literature on the associations between patient and physician characteristics and use of low tidal volume ventilation in ARDS. We confirm prior findings that hypoxemia severity and ARDS documentation are associated with a higher likelihood of LTVV use. We demonstrate the novel result that even after normalizing for height, there is still a relationship between height and a patient’s likelihood of receiving LTVV, suggesting that the association moves beyond the height-dependent nature of LTVV threshold calculations. We conclude that when evaluating LTVV utilization, the height of the patient population should be accounted for. 

 Second, we construct a model for physician recognition of ARDS that accounts for hypoxemia, patient height, and ARDS documentation, which is novel in the literature. The model requires data that can easily be mined from an electronic health record, making it feasible at resource-constricted institutions, where other methods – such as surveys – would not be possible. This model can be integrated into a decision support or audit-and-feedback system for more continual assessment and intervention.

We thank you for your work and feedback.

---

## [Decision Letter · Decision Letter 1]

10 Sep 2019

[EXSCINDED]

A quantitative approach for the analysis of clinician recognition of acute respiratory distress syndrome using electronic health record data

PONE-D-19-17952R1

Dear Dr. Bechel,

We are pleased to inform you that your manuscript has been judged scientifically suitable for publication and will be formally accepted for publication once it complies with all outstanding technical requirements.

With kind regards,

Lars-Peter Kamolz, M.D., Ph.D., M.Sc.

Academic Editor

PLOS ONE

Additional Editor Comments (optional):

Reviewers' comments:

Reviewer's Responses to Questions

**Comments to the Author**

1. If the authors have adequately addressed your comments raised in a previous round of review and you feel that this manuscript is now acceptable for publication, you may indicate that here to bypass the “Comments to the Author” section, enter your conflict of interest statement in the “Confidential to Editor” section, and submit your "Accept" recommendation.

Reviewer #1: (No Response)

Reviewer #2: All comments have been addressed

2. Is the manuscript technically sound, and do the data support the conclusions?

Reviewer #1: (No Response)

Reviewer #2: Yes

3. Has the statistical analysis been performed appropriately and rigorously? 

Reviewer #1: (No Response)

Reviewer #2: N/A

4. Have the authors made all data underlying the findings in their manuscript fully available?

Reviewer #1: (No Response)

Reviewer #2: Yes

5. Is the manuscript presented in an intelligible fashion and written in standard English?

Reviewer #1: (No Response)

Reviewer #2: Yes

6. Review Comments to the Author

Reviewer #1: (No Response)

Reviewer #2: The authors have responded accordingly to the editor´s and author´s comments.

7. PLOS authors have the option to publish the peer review history of their article (what does this mean?). If published, this will include your full peer review and any attached files.

Reviewer #1: No

Reviewer #2: No

---

## [Editor Report · Acceptance letter]

13 Sep 2019

PONE-D-19-17952R1 

A quantitative approach for the analysis of clinician recognition of acute respiratory distress syndrome using electronic health record data 

Dear Dr. Bechel:

I am pleased to inform you that your manuscript has been deemed suitable for publication in PLOS ONE. Congratulations! Your manuscript is now with our production department. 

With kind regards,

on behalf of

Dr. Lars-Peter Kamolz 

Academic Editor

PLOS ONE